# Hybrid Bag-of-Visual-Words and FeatureWiz Selection for Content-Based Visual Information Retrieval

**DOI:** 10.3390/s23031653

**Published:** 2023-02-02

**Authors:** Samy Bakheet, Ayoub Al-Hamadi, Emadeldeen Soliman, Mohamed Heshmat

**Affiliations:** 1Faculty of Computers and Artificial Intelligence, Sohag University, Sohag 82524, Egypt; 2Institute for Information Technology and Communications (IIKT), Otto-von-Guericke-University Magdeburg, 39106 Magdeburg, Germany

**Keywords:** CBIR, BoVW, RootSIFT, information retrieval, visual words, FeatureWiz, SURF and FAST detectors, SVMs, Corel-1000, Caltech-10, Oxford Flower-17

## Abstract

Recently, content-based image retrieval (CBIR) based on bag-of-visual-words (BoVW) model has been one of the most promising and increasingly active research areas. In this paper, we propose a new CBIR framework based on the visual words fusion of multiple feature descriptors to achieve an improved retrieval performance, where interest points are separately extracted from an image using features from accelerated segment test (FAST) and speeded-up robust features (SURF). The extracted keypoints are then fused together in a single keypoint feature vector and the improved RootSIFT algorithm is applied to describe the region surrounding each keypoint. Afterward, the FeatureWiz algorithm is employed to reduce features and select the best features for the BoVW learning model. To create the codebook, K-means clustering is applied to quantize visual features into a smaller set of visual words. Finally, the feature vectors extracted from the BoVW model are fed into a support vector machines (SVMs) classifier for image retrieval. An inverted index technique based on cosine distance metric is applied to sort the retrieved images to the similarity of the query image. Experiments on three benchmark datasets (Corel-1000, Caltech-10 and Oxford Flower-17) show that the presented CBIR technique can deliver comparable results to other state-of-the-art techniques, by achieving average accuracies of 92.94%, 98.40% and 84.94% on these datasets, respectively.

## 1. Introduction

An image is more than thousands of words and is the cornerstone of visual media representation. With the tremendous development in information technology, computer engineering, and smartphone, the number of shared images over networks has increased rapidly [1,2]. Thus, the world needs a robotic way to access and represent visual data concerning content rather than meta-data.

Computer interpretation of the contents of an image is less trivial. A machine’s vision is an extensive matrix of numbers, and it has no idea of the thoughts, knowledge, or meaning of the image it tries to convey. To understand the contents of an image, apply image classification and retrieval that utilize machine learning algorithms’ tasks to extract meaning from an image. This process could be as simple as assigning a label to what the image contains. Searching for an image based on its content or example is known as CBIR or Query By Image Content (QBIC). Many modern applications in hospitals, factories, geographical information systems, remote sensing, cultural heritage, mining, education, home entertainment, journalism and advertising, fashion and interior design, crime prevention, military, medical diagnosis, web searching, etc., use CBIR models to make their systems more intelligent and robust [3].

The first search by an image system was built by IBM [4]. After that, CBIR gained acceptance in many areas of computer vision, becoming crucial in producing robust techniques. Many research papers overviewed CBIR [5,6,7,8,9]. Recently, the image features have been based on color, texture, and shape to quantify image content [10,11,12]. Then, various algorithms were developed to describe the features of the image. At the highest level, a feature is the interesting regions of an image that are both unique and recognizable. The process of finding and describing the interesting regions of an image is known as feature extraction, and it is broken down into two phases: Keypoint Feature Detection (KFD) and Feature Description (FD). The first phase is to detect points that determine the interesting regions of an image. These regions may be edges, corners, or blobs where the point intensities are approximately uniform. The algorithms that can detect these interesting regions are Keypoint Feature Detectors (KFDs). The second phase is to describe and quantify the region of the image surrounding each interesting keypoint. To describe a region of an image, extract the features vectors of each keypoint called a local feature. Only the local neighborhood surrounding the keypoint is included in the computation of the descriptor. The algorithms that apply this phase are called FDs.

Feature-level fusion is a critical stage in feature preparation. The advantage of feature fusion lies in two aspects: first, it can derive the most discriminatory information from the original multiple feature sets involved in fusion; second, it can eliminate redundant information resulting from feature set correlation, allowing for real-time decision-making. In other words, feature fusion can derive and obtain the most effective and low-dimensional feature vector sets that will help with the final conclusion. Although the merits and importance of feature fusion in the CBIR applications, it has some challenges where it must run some experiments on the dataset to select the most appropriate features to fuse between them. Moreover, it should tune the appropriate percentage of features which will be used in the learning stage to avoid the overfitting of data in the learning stage. The feature fusion between the KFDs introduced in this paper is more effective than the recent fusion techniques between the FDs, where the extracted features at the fusion between KFDs are less than in the case of the fusion between FDs, which decreases the probability of overfitting and presenting more accurate results. Furthermore, the KFDs fusion has less computational complexity than the FDs fusion.

There are many techniques for CBIR. However, the outcomes techniques obtained using a single feature type are unsatisfactory. Multiple combined feature descriptors are frequently used to achieve better outcomes, but quickly and effectively searching a database for a relevant query image is still challenging. Moreover, the computational complexity is higher for the existing CBIR systems. To overcome the challenges mentioned above, besides giving better results; this paper introduced four main contributions to improving the CBIR systems, first, comparing the strongest nine KFDs parallel with the seven robust FDs. The comparison is to select the best two KFDs with the most convenient FDs. This contribution produces effective low-dimensional Hybrid Keypoint Features Vector (HKFV) that reduce the next steps’ complexity, cost, and time. The second contribution is using a new feature selection algorithm (i.e., FeatureWiz selection) to select the powerful features from the extracted features for the next clustering step. This contribution determines the overall powerful features extracted from the dataset that reduces the time complexity and cost for the clustering stage. Furthermore, it decreases the data overfitting probability at the learning stage. The third contribution is applying a feature selection algorithm on the BoVW to select adequate visual words in the learning and sorting stage, called BoVW FeatureWiz Selection (BFS). This contribution improves the learning stage and decreases the data overfitting probability. It also contributes to reducing the time complexity and cost of similarity matching and sorting. The fourth contribution is the new Keypoint Feature Fusion Visual Words (KFFVW) framework. The rest of the paper is structured as follows: Section 2 reviews the literature related to CBIR. Section 3 discusses in detail the pipeline of the proposed adaptive KFFVW framework, whereas the experimental results are illustrated and evaluated in Section 4. Section 5 concludes the research paper.

## 2. Related Work

CBIR has been an active growing research area for the last three decades [13]. Feature extraction is the baseline that quantifies a dataset in CBIR by extracting features from each image in the collection. The feature extraction step is the foundation of adaptive image classification, and its success influences the classification performance. Feature extraction is heavily used in computer vision applications to rank the similarity between images. The feature extraction technique may be carried out automatically in the CBIR, significantly reducing the challenges of the pure annotation-based approach in Text-Based Image Retrieval (TBIR). Nilsback et al. [14] proposed a visual vocabulary that explicitly represents color, texture, and shape to distinguish one flower from another in 17 flower species. Latha et al. [15] introduced a new color-image-descriptor through feature fusion between HSV color statistical feature and the multichannel LBP-oriented color descriptor for an enhanced hybrid CBIR system. Sadek et al. [16,17] proposed an architecture of the CBIR system using the neural spline model to employ a cubic-spline activation function to reduce the semantic gap between the high-level semantic human vision concepts and low-level visual features. Specific texture approaches, such as the Gabor features, the Curvelet, and the Wavelet packets, are used in CBIR to reduce the semantic gap [18,19,20].

Ghahremani et al. [21] proposed an image feature integration technique based on the Local Binary Pattern (LBP) descriptor and the Scale Invariant Feature Transform descriptor (SIFT) as a local invariant descriptor, and the Histogram of Oriented Gradients (HOG) as a global feature to produce the Bag-of-Features (BoF). The SIFT and LBP detect feature keypoints, while the HOG produces powerful features for the background clutter. The results of [21] showed that the most accurate CBIR was achieved using a combination of SIFT and LBP.

The BoVW is a common effective technique in image retrieval tasks. Yousuf et al. [22] proposed a novel method based on the SIFT features and a new weighting scheme. A fuzzy representation was used to index images with a more stable signature in the used method. Alkhawlani et al. [23] used the SURF and the SIFT as local invariant descriptors and utilized k-means for clustering the features. Support Vector Machines (SVMs) were used as an effective classifier for the learning and classification stage. Andy et al. [24] used the Gaussian Mixture Model (GMM) to construct a Visual Words Codebook (VWC). Moreover, multiclass SVMs were proposed to classify images using Hellinger’s kernel, chi-square kernel, and linear kernel functions for similarity metrics. The color histograms of the query image and images of the same class were used to compute similarity distance. Mehmood et al. [25] proposed a novel Weighted Average of Triangular Histograms (WATH) technique to improve the performance of visual vocabulary in BoVW. If the vocabulary size is too small, it will not represent all image patches, while the vocabulary size is too large, the overfitting issue will arise. The WATH technique decreases this issue on the larger size of the codebook.

Khan et al. [26] proposed a feature fusion vector by merging Local Tetra Angle Patterns and RGB color features. They also optimized SVMs using Genetic Algorithms (GA) for better image classification. They used the Chi-square quadratic distance function to measure the similarity between the query image and the dataset images. Alarcao et al. [27] proposed an Expert, dynamic Late Feature (ExpertosLF) fusion system for CBIR based on online learning via end-user relevance feedback. At each query, ExpertosLF determines the contribution of each feature collection in the ensemble for the subsequent queries based on the advantage of the user’s feedback.

## 3. The Proposed Methodology

Lastly, CBIR has many techniques, but each has its specific process. This work focused on the recent techniques that depend on BoVW. The latest fusion techniques in BoVW considered the fusion between feature descriptors. The suggested method improves the extracted features by fusing the keypoint features, and it is then described by feature descriptor algorithms rather than the direct fusion between the feature descriptors.

The proposed method depends on extracting the FAST and the SURF keypoints separately from each image. Then, these keypoints are fused in a single keypoint feature vector. The RootSIFT algorithm is applied to describe the region surrounding each keypoint. A large number of feature vectors characterizes each image. Afterwards, a feature selection algorithm is used to select the appropriate feature vectors from all images as a sample for the clustering step. A k-means clustering algorithm is applied to formulate feature vectors into clusters. Each cluster is a group of feature vectors whose center represents a visual world. The combination of the visual words constructs the codebook. Then, vector quantization is applied to this codebook to describe each image in one dimension feature vector. The output of the vector quantization is a separate feature vector for each image in the training dataset.

The query process requires comparing the query features to every feature vector in the dataset. It is a linear operation O(N). As the size of the dataset increases, the time it takes to perform a query will increase. Building an inverted index of the BoVW to optimize the data structures of the indexed features breaks down the query process to sub-linear O(log(N)). Figure 1 shows the block diagram of the proposed method.

### 3.1. Feature Extraction

Before the feature extraction phase, a preprocessing step is performed for computational efficiency. The images in the collection are first converted to HSV-color format and then down-sampled to a fixed resolution of 320 × 320 pixels. Feature extraction in CBIR is the process of representing a broad set of images by applying an adequate image descriptor to the keypoints of images to extract features from each image in the dataset. One of the most influencing factors of the BoVW is the feature extraction process, which contributes to constructing powerful visual words. Keypoints detectors fusion in this paper contributes to the success of the feature extraction process for a better BoVW. The feature extraction stage is the baseline of the adaptive image classification, and its success determines the success of the overall classification performance [28]. After running sufficient experiments and validations, the FAST and SURF keypoint detectors were selected to detect the interest points in an image. The Root SIFT descriptor was chosen to describe those points.

#### 3.1.1. Speeded-Up Robust Features (SURF)

Bay et al. [29] provided a quick and reliable SURF algorithm for local, similarity invariant representation and comparison of images. The SURF algorithm key is its ability to compute operators quickly using box filters, enabling real-time applications such as CBIR, tracking, and object detection. SURF is composed of a two-step Fast Hessian keypoint detector and feature description. In this paper, a single first step is utilized to identify keypoints (i.e., points of interest) in an image at locations where the determinant of the Hessian matrix has a maximum value. In the second step, feature description is implemented using the RootSIFT algorithm. In the first step, the Hessian matrix performs well in calculation speed and accuracy, which is why SURF utilizes it. The Hessian of a given pixel at point X = (x,y) is defined as follows:(1)$$H\left(f\left(x,y\right)\right)=\left[\begin{array}{cc} \frac{\partial^{2}f}{\partial x^{2}} & \frac{\partial^{2}f}{\partial x\partial y}\\ \frac{\partial^{2}f}{\partial x\partial y} & \frac{\partial^{2}f}{\partial y^{2}} \end{array}\right]$$

For adapting to any scale, the image is filtered by a Gaussian kernel. The Hessian matrix H (x, σ) in a given image I can be defined at a position *x* and scale σ as follows:(2)$$H\left(f\left(x,y\right)\right)=\left[\begin{array}{cc} L_{xx}\left(x,\sigma \right) & L_{xy}\left(x,\sigma \right)\\ L_{xy}\left(x,\sigma \right) & L_{yy}\left(x,\sigma \right) \end{array}\right]$$
where,

$L_{xx}\left(x,\sigma \right)=I\left(x\right)*\frac{\partial^{2}}{\partial x^{2}}g\left(\sigma \right),L_{xy}\left(x,\sigma \right)=I\left(x\right)*\frac{\partial^{2}}{\partial x\partial y}g\left(\sigma \right),L_{yy}\left(x,\sigma \right)=I\left(x\right)*\frac{\partial^{2}}{\partial y^{2}}g\left(\sigma \right)$ where $L_{xx}\left(x,\sigma \right)$ is the convolution of the Gaussian second-order derivative with the image I in point x, and similarly for $L_{xy}\left(x,\sigma \right)$ and $L_{yy}\left(x,\sigma \right)$, g(σ) as scale σ. The discretized second-order Gaussian derivatives in the yy and y directions are approximated by weighted box filters in Figure 2.

To quickly and accurately determine the sum of values in a given image or a rectangular portion of a grid, utilize the integral image. It can be used to determine the average intensity inside a particular image. They make it possible to compute box-type convolution filters quickly. The total pixels in the input image I inside the rectangular region bounded by the origin and *x* is represented by the entry of an integral image $I_{sum(x)}$ at a location x=(x,y) is given by:(3)$$I_{\sum (x)}=\sum_{i=0}^{i\leq x}\sum_{j=0}^{j\leq x}I\left(i,j\right)$$

Convolution with a Gaussian kernel must be used to determine the Hessian matrix’s determinant and the second-order derivative. SURF uses box filters to enhance the approximation (i.e., convolution and second-order derivative). These approximative second-order Gaussian derivatives can be calculated using an integral image regardless of size at a much lower computational cost, which is a part of why SURF is fast. Applying box filters of various sizes in SURF allows for scale-space implementation. Therefore, rather than iteratively shrinking the image size, the scale space is utilized by upscaling the filter size. A 9 × 9 size filter with scale s = 1.2 (i.e., equivalent to Gaussian derivatives with σ = 1.2) serves as the box filter’s initial scale layer. The subsequent layers are created by filtering the image with progressively larger masks while accounting for integral images’ discrete character and their unique filter structure. This yield filters of sizes 9 × 9, 15 × 15, 21 × 21, and 27 × 27.

The Non-Maximum Suppression (NMS) technique in a 3 × 3 × 3 neighbourhood is used to localize interest points in the image. The Brown et al. approach [30] is then used to interpolate the maxima of the determinant of the Hessian matrix in scale and image space. The scale difference between each octave’s first layers is more extensive, making scale space interpolation crucial. So the NMS technique can be used to locate the keypoints over an image scale. For achieving a good approximation, the determinant is weighted to exclude keypoints having low contrast or points lying on edges or being near to edges:(4)$$det\left(H_{approx}\right)=D_{xx}D_{yy}-{\left(\omega D_{xy}\right)}^{2}$$
where $D_{yy}$ and $D_{xy}$ are the approximated and discrete kernels for $L_{yy}$ and $L_{xy}$, respectively. The ω term is theoretically sensitive to scale, but it can be assumed almost constant at 0.9 as Bay [29] suggested.

#### 3.1.2. Features from Accelerated Segment Test (FAST)

The FAST idea is based on the corner detector known as the Smallest Univalue Segment Assimilating Nucleus (SUSAN). The brighter and darker pixels next to each other are determined using the center of a circular area. Only the pixels on the discretized circle that characterize the segment, not the entire circle, are evaluated. Sixteen pixels must be compared to the nucleus value for a full accelerated segment test as shown in Figure 3. Therefore, the value S defines the detected corner’s greatest angle. The corner detector is more repeatable when S is kept as high as possible. When comparing the brightness of the nucleus with the value of a pixel on the circular pattern, the Accelerated Segment Test (AST) uses a minimal difference threshold (t). This setting controls the sensitivity of the corner reaction. A lower t-value produces corners with smoother gradients which are fewer but stronger corners, whereas a big t-value only produces weak corners. When comparing FAST in [31,32] to other corner detectors such as the Harris detector (HARRIS), Difference of Gaussian (DoG), or SUSAN, it is demonstrated that the AST with S = 9 or 16 has good repeatability. A corner detector’s repeatability is a quality criterion that assesses a method’s ability to identify similar corners of a scene from various points.

#### 3.1.3. Hybrid Keypoint Features Vector (HKFV)

As shown in Table 1, Table 2 and Table 3 the feature extraction process requires a longer time and higher hardware than other CBIR processes. Increasing the number of features will increase the time needed for other steps depending on the number of features in this step. Therefore, avoiding expanding the number of feature detectors or feature descriptors in the feature fusion stage. In our work, the output keypoints of each FAST and SURF were concatenated, only including the unique key points to develop the HKFV. It is essential to mention that FAST or SURF alone cannot be better keypoint features for images of similar color, texture, or objects as the HKFV. Each dataset image has a single HKFV that represents the interesting salient regions of an image. The images of a dataset do not have the same length for HKFV, but the output of describing each keypoint in HKFV is a feature vector of the same dimension. The selection of (FAST + SURF) feature detectors incorporated in our work is based on two main reasons, scientific explanation and experimental results.

Firstly, the scientific explanation. In many computer vision tasks, the FAST is a corner-detection technique that might extract feature points and then track and map objects. The FAST corner detector’s high computing efficiency may be its most advantageous feature. Furthermore, using machine learning techniques, faster and more efficient computations may be made by computer vision tasks using FAST detectors. The FAST detector is quicker than the difference of Gaussians (DoG), employed by the SIFT, SUSAN, Harris detectors, and many other popular feature extraction techniques. In addition, FAST extracts more keypoints than other detectors. Additionally, when machine learning techniques are used, the outcomes are better.

The SURF method is a quick and reliable approach for local, similarity-invariant representation and comparison of images. The SURF approach’s primary appeal is its ability to compute operators quickly using box filters, opening the door to real-time applications, including image matching, object identification, image registration, and classification. The SIFT feature detector stage served as some of the inspiration for SURF authors. The authors of SURF say that the standard version is more resistant to various image alterations than SIFT and is many times quicker than SIFT.

On the objective of feature detection, ORB outperforms SIFT and is better than SURF while being nearly two orders of magnitude quicker. The well-known FAST keypoint detector and the BRIEF description are the foundation for ORB. Both methods are appealing due to their high effectiveness and low expense.

Secondly, the experimental results. The experimental comparisons of the most vital nine Keypoint Feature Detectors (KFDs) parallel with the seven robust Feature Descriptors (FDs) are implemented based on the BOVW technique, as shown in Figure 4. These results indicate that the top two keypoint detectors are FAST and SURF. Furthermore, the Fast or SURF outperforms the ORB feature detectors.

Based on the previous scientific explanation and the experimental results, it was very interesting to find results of the best combination between SIFT, FAST, SURF, and ORB to produce the HKFV, but the experimental results shown in Table 3 indicate that the combination FAST+SURF gives better results and time performance than SIFT+ORB or SIFT+ORB+FAST+SURF. That is because the BOVW technique depends not only on the image keypoints and image features but also on the image patches that represent the visual words.

To form the HKFV of FAST+SURF, Let Image I is represented as:(5)I=(P(x,y))
where P(x,y) represents a pixel at the position (x,y). The FAST keypoints (F) and The SURF keypoints (S) are detected from the image and are represented as
(6)$$F=\{f_{1},\;f_{2},\;\ldots ,\;f_{i}\}$$
(7)$$S=\{s_{1},\;s_{2},\;\ldots ,\;s_{j}\}$$
where $f_{1}$ to $f_{i}$ and $s_{1}$ to $s_{j}$ represent the keypoints detected by FAST and SURF, respectively. While *i* and *j* are the numbers of those keypoints. To form the unique keypoints ($FS_{k}$) of the two sets Fk and Sk find the union between them
(8)FS=F⋃S
so the set of FS ${}_{k}$ keypoints could be represented as
(9)$$FS=\{FS_{1},\;FS_{2},\;\ldots ,\;FS_{k}\}$$
where *k* represents the number of unique keypoints of FAST and SURF.

#### 3.1.4. Scale Invariant Feature Transform Descriptor (SIFT)

The SIFT feature descriptor algorithm requires a set of input keypoints. Then, for each of the input keypoints, SIFT takes the 16 × 16 pixel region surrounding the center pixel of the keypoint region, then divide the 16 × 16 pixel region into sixteen 4 × 4 pixel windows. For each of the 16 windows, compute the gradient magnitude and orientation. After that, construct an 8-bin histogram for each 4 × 4 pixel window. Then collect all 16 of these 8-bin orientation histograms and concatenate them to produce a feature vector of 16 × 8 = 128 dimension. It was, lastly, normalizing the entire feature vector.

#### 3.1.5. RootSIFT Descriptor

There are two stages to local feature descriptions. In the first stage, intriguing and conspicuous points of an image need to be detected. These points, known as keypoints, might be edges or corners in an image. This stage is determined using HKFV. The region surrounding each keypoint in an image must first be extracted and quantified once the collection of keypoints has been determined. Since only the local neighborhood region around the keypoint is considered in the descriptor computation, the feature vector associated with a keypoint is referred to as a feature or local feature. It must select a robust feature descriptor algorithm to quantify and describe the surrounding region for each keypoint in HKFV. Arandjelovic et al. [33] introduced a simple extension to SIFT called RootSIFT, which can be used to increase object recognition accuracy, quantization, and retrieval accuracy. The following simple steps are to extend SIFT to RootSIFT.

(i)Compute the SIFT descriptors.(ii)L1-normalize each SIFT vector.(iii)Take the square root of each element in the SIFT vector.

In our work, we decided on the RootSIFT feature descriptor for this task. This selection was based on experimental reasons. The experimental comparisons of the most vital nine KFDs parallel with the seven robust FDs shown in Figure 4 indicate that the RootSIFT descriptor gives the highest results with each FAST and SURF detector.

### 3.2. FeatureWiz Selection

When building predictive models, feature variables are crucial. Many features are undesirable since they could produce overfitting, which would cause the model to resemble the training data closely. Additionally, a lot of characteristics will increase the size of the search space for the problem, which is known as the curse of features. A selection features approach could be employed to give a relevant score for each feature variable and determines which features are essential for predicting the target variable. The feature selection model was down-sampled into two levels. In the first one, the algorithm selects the unique features, approximately 25% to 50% of the extracted features. The second level uses the open-source Python package FeatureWiz, a quick and effective technique to identify essential features in a dataset regarding the target variable. The FeatureWiz processes operate in two stages. The first stage is the Searching for the Uncorrelated List of Variables (SULoV), which identifies a pair of variables beyond an externally passed correlation threshold, so they are highly correlated. After determining the pairs, the Mutual Information Score (MIS), which gauges the relationship between two variables, is calculated for each pair. The pair of variables with the lowest correlation and the highest MIS scores are then considered. The variables chosen from SULOV are recursively sent through XGboost in the second stage (Recursive XGBoost), which aids in identifying the best features by skipping the most similar data. One of the optional FeatureWiz parameters is the percentage of the output features from the total input features. The FeatureWiz Percentage Parameters (FPP) will be tuned based on various experiments to select the feature percentage that gives the best retrieval results. The chosen features percentage for the clustering could be calculated as
(10)$$R_{R}=C*R_{D}$$
where $R_{R}$ represents the selected features, *C* is the value of the percentage (0 ⩽C⩽ 1), and $R_{D}$ is the total extracted features.

### 3.3. Clustering to Visual Words

It is common to group features in an image to give them meaning. K-Means clustering is the most common clustering algorithm for this grouping. Clustering is splitting a large number of data points into a small number of clusters. Data points are assigned to the closest centroid to build a cluster. Therefore, data points in one cluster are easier to compare with data points in other clusters. The center of each cluster is called a visual word. The combination of visual words constructs the vocabulary. The following Figure 5 shows the K-Means clustering model. After K-Means clustering is applied, the output visual words vocabulary could be denoted as follows:(11)$$\left\{R_{R}\right\}\overset{K-Means}{\to }VW=\left\{vw_{1},vw_{2},\ldots ,vw_{c}\right\}$$
where $vw_{1}$ to $vw_{c}$ are visual words feature vectors.

### 3.4. Features Quantization

To quantize feature vectors of an image to a single BoVW histogram vector, the Cosine distance matrix D between the total extracted features $R_{D}$ for each image and the visual words vector VW must be constructed. The CDM between the vectors *F* and *W* is calculated as follows:(12)$$\begin{array}{c} d_{F,W}=Cos\left(F,W\right)=\frac{F\cdot W}{\left\|F\|\;\|W\right\|} \end{array}$$

The Cosine distances matrix D for $\left(f_{1},f_{2},\ldots ,f_{c}\right)$ in the total extracted features matrix $R_{D}$ and $\left(w_{1},w_{2},\ldots ,w_{z}\right)$ in the visual words VW is calculated as follows:(13)$$\begin{array}{c} D\left(R_{D},VW\right)=\left[\begin{array}{cccc} d_{f_{1}w_{1}} & d_{f_{1}w_{2}} & \cdots & d_{f_{1}w_{c}}\\ d_{f_{2}w_{1}} & d_{f_{2}w_{1}} & \cdots & d_{f_{2}w_{c}}\\ \vdots & \vdots & \ddots & \vdots \\ d_{f_{z}w_{1}} & d_{f_{z}w_{1}} & \cdots & d_{f_{z}w_{c}} \end{array}\right] \end{array}$$

Then apply image mapping by forming the histogram that describes the image using VW. The histogram is constructed by counting how many each visual word gives the minimum distance d in matrix D. It is calculated as:(14)$$h\left(w_{i}\right)=count\left(argmin\left(row_{k}\right)\;is\;i\right)\;for\;k\in \left(1,z\right)\;and\;i\in \left(1,c\right)$$
where argmin returns the index of the column of the minimum value in the row $row_{k}$ in matrix D. Therefore, the output histogram of an image is:(15)$$H=\{h\left(w_{1}\right),h\left(w_{2}\right),\ldots ,h\left(w_{c}\right)\}$$

### 3.5. Visual Words Features Normalization

A scaling technique called normalization shifts and rescales values so that they fall between the ranges of 0 and 1. They are additionally called Min-Max scaling. The normalization step reduces the overfitting probability. The normalization process could be calculated by:(16)$$\overset{`}{H}=\frac{H-h_{min}}{h_{max}-h_{min}}$$
where $h_{min}$ and $h_{min}$ are the minimum and the maximum features of the H visual words features vector respectively.

### 3.6. BoVW FeatureWiz Selection (BFS)

The Bag-of-Words (BoW) model represents text data at modelling text using machine learning algorithms. The BoW model is a well-known model for feature extraction in natural language processing. The BoW is a representation of text that describes the occurrence of words within a document. Selecting powerful words from a text is a crucial step for the success of the BoW model. In Visual retrieval, there is a similar model, which is the BoVW model, and it operates as same as the BoW model technique. In our work, we added a step for selecting powerful visual words after clustering. This step is built based on the FeatureWiz selection technique. FeatureWiz is used for the feature selection before clustering and to choose the most powerful visual words after the normalization step.

### 3.7. The Support Vector Machines (SVMs) Prediction Model

The prediction model is applied using the state-of-the-art SVMs on the BoVW. The SVMs are initially proposed by Vapnik [34]. The SVMs are a supervised machine learning algorithms that are used for regression or classification. Although regression can also benefit greatly from it, classification is where it is most often used. The SVMs algorithm seeks to identify a hyperplane in a d-dimensional space that clearly divides the data points into various classes according to the support vectors. Support vectors in both classes are the data points that are closest to the hyperplane. In a d-dimensional space, where d is the number of features in the dataset, each individual piece of data was plotted. The best hyperplane to divide the data should then be determined. SVMs are limited to binary classification (i.e., choose between two classes). For multi-class issues, though. Each class can have a binary classifier created by the SVMs. Each classifier will produce one of two outcomes based on a score value: either the data point belongs to that class or does not. The final output of the SVMs is the class with the highest score. Consider the case where SVM will divide the two classes, A and B, in a two-dimensional space. The classifier’s goal is to determine if the unidentified feature vector F belongs to Class A or Class B. Suppose a linear equation:(17)$$g\left(F\right)=\omega^{T}F+\theta$$
where ω denotes the weight vector’s perpendicular position to the hyperplane. It represents the hyperplane’s orientation in d-dimensional space, while θ shows where the hyperplane is located. This linear equation depicts a straight line in two dimensions. In contrast, the equation represents a plane in three dimensions and a hyperplane in more than three dimensions. Figure 6 shows the SVM in two-dimensional space.

If $g(f_{i})<-1$ then, $f_{i}$ lies in class B.If $g(f_{i})=-1$ then, $f_{i}$ is a positive hyperplane.If $g(f_{i})=0$ then, $f_{i}$ lies on the hyperplane.If $g(f_{i})=1$ then, $f_{i}$ is a negative hyperplane.If $g(f_{i})>1$ then, $f_{i}$ lies in class A.

The mathematical description of SVM has several approaches, such as optimal separating hyperplanes, linearly separable, linearly non-separable, and nonlinear SVM. In the proposed work, it is recommended to use linear SVM duo to many features as it is more likely that the data are linearly separable in high dimensional space. In Linear SVM, we want only to tune one parameter (c), which controls the amount of regularization applied to the data. A smaller value of C allows the SVM to be a soft classifier. As the value grows, the SVM becomes more and more strict, allowing for fewer misclassifications; however, if it grows too large, it could easily overfit the training data.

## 4. Experimental Results

This section illustrates the outcomes of the experiments and the discussion of the proposed KFFVW framework. The performance and accuracy are evaluated on the three standard datasets (i.e., Corel-1000, Caltech-10, and Oxford Flower-17). The experiments were focused on the Corel-1000 dataset to select the best two keypoints detectors to apply fusion between them to form robust HKFV for the features’ extraction stage. The FAST and SURF keypoint feature detectors gave the best performance. The performances of each standalone FAST and SURF keypoint feature detector based on traditional CBIR were evaluated respectively on various vocabulary sizes compared to the performance of the proposed HKFV with the BFS for the proposed KFFVW framework. Every experiment was implemented five times based on K-fold Cross-Validation (KCV) to achieve the average performance. Furthermore, the performance of the proposed KFFVW was compared to the recent state-of-the-art methods on the specified datasets.

### 4.1. Experimental Parameters

The CBIR parameters are a primary factor for evaluating any CBIR system’s performance and running time. The parameters of the different experiments for the proposed framework are detailed below.

#### 4.1.1. Features Percentages

The extracted features were used to construct the vocabulary. Using all the extracted features as input to the clustering step is not recommended. Datta et al. [35] reported that, although the increase in the used features gives better results, it increases the processing cost and tends to overfit and vice versa. Every used vocabulary size was tested over various feature percentages (i.e., 10%, 25%, 50%, 75%, and 100%) of the total unique extracted features to find the best percentage features that give the best performance with minimum processing cost. The features were selected based on featureWiz selection.

#### 4.1.2. Vocabulary Size

The vocabulary size is the number of feature clusters. The vocabulary size is a primary key for evaluating any CBIR System. More extensive size vocabulary increases the system’s performance, but the massive vocabulary size tends to overfit and the processing cost and vice versa [36]. Various vocabulary sizes (i.e., 256, 512, 1024, …, 32,780) were examined to select the most suitable vocabulary size from the training features for efficient vector quantization.

#### 4.1.3. Training and Testing Image Percentage

The experimented datasets are designed in two groups; one is for testing, and the other is for training. The training group is selected based on five-fold cross-validation. The dataset is divided into 80% for the training process and 20% for the testing stage.

### 4.2. Evaluation Metrics

The KCV is a simple approach for selecting the best model and determining an estimator’s accuracy [37]. The primary parameter for measuring the performance of the suggested CBIR framework is KCV. According to [38], the best value for k is between 5 and 10. The performance of the framework is evaluated using five-fold cross-validation. Precision is measured and summarized by each dataset class. The overall precision of the CBIR system is calculated as the average precision in all classes. As we set the value of k to 5, 80% of data will always be utilized to develop the training model, while the remaining 20% will be used to validate the model. The validation process was repeated five times, with the average being computed (each time was for a different fold). Various evaluation measures are used, including accuracy, precision, MAP, recall, and F1-score, which can be stated in the following ways.

*Accuracy:* Accuracy relates to how close a measurement is to the true or recognized value. That means it is the probability that the CBIR framework test retrieves the correct images. It is one of the most intuitive and widely used performance measures.
(18)$$Accuracy=\frac{TN+TP}{FN+FP+TN+TP}\times 100\%$$*Precision (P):* Precision relates to the capability to detect the true class among all predicted valid classes accurately. Moreover, it is expressed as a ratio of the correctly predicted valid classes to all anticipated true classes. High precision is always associated with a low false positive rate, as shown by the equation below.
(19)$$P=\frac{TP}{FP+TP}\times 100\%$$*Mean Average Precision (MAP):* Often known as the average precision (AP), it is a well indicator for assessing how well models execute tasks such as document/information retrieval and object recognition. The MAP of a set of queries is defined by
(20)$$MAP=\frac{\sum_{q=1}^{Q}AveP\left(q\right)}{Q}\times 100\%$$
where AveP(q) is the average precision (AP) for a specific query, q, and Q is the total number of queries in the collection.*Recall (R):* Recall, also known as sensitivity, hit rate, or true positive rate, can be broadly considered as the ability of the model to identify all positive cases so that it can be expressed as
(21)$$R=\frac{TP}{FN+TP}\times 100\%$$It should be noted that the above formula shows that high recalls are certainly associated with low false negative rates.*F1-score (F):* F1-score score is not as intuitive as accuracy, but it is beneficial when measuring the robustness and accuracy of a CBIR system. The F1-score, which serves as a single measure of test performance that describes both recall and precision, is usually calculated as a weighted average of recall and precision.
(22)$$F=2\frac{R\cdot P}{R+P}\times 100\%$$

### 4.3. Datasets

The Corel-1000 dataset [39] is a subset of the Corel image dataset [40], which is common in the evaluation of the CBIR [41,42,43]. It includes 1000 images in Corel-1000, divided into ten semantic classes having 100 images for each. Figure 7 shows a sample image of each semantic class of the Corel-1000 dataset. The Caltech-10 dataset is a subset of the Caltech-101 dataset [44]. The Caltech-10 includes 1000 images divided into ten classes, each with 100 images. The Oxford Flower-17 dataset [14] contains 1360 images divided into 17 semantic flower classes, where each class consists of 80 images. Figure 8 and Figure 9 show a sample image of each semantic class of the Oxford Flower-17 and Caltech-10 datasets, respectively.

#### 4.3.1. Corel-1000 Dataset Results

For selecting the best keypoint detectors with the best features descriptor, each keypoint detector of the (FAST, HARRIS, Good Features To Track (GFTT), DoG, SURF, CenSurE, Maximally Stable Extremal Regions (MSER), Binary Robust Invariant Scalable Keypoint (BRISK), and Oriented FAST and Rotated Binary robust independent elementary features (ORB)) with each features descriptor of the (RootSIFT, SIFT, SURF as a descriptor, BRISK as a descriptor, ORB as a descriptor, Fast Retina Keypoint (FREAK), and Binary Robust Independent Elementary Features (BRIEF) have experimented.

The experiments were on the Corel-1000 dataset, where 2048 clusters were used to form the vocabulary set. Each experiment was tested five times with a different collection of images for the training (80%) and the rest for testing. Then, the average of all results was calculated. The performance comparison in Figure 4 shows that the two maximum MAP values on two different keypoint detectors are 82.76% at using FAST as a keypoint detector and RootSIFT as a features descriptor and 77.47% at using SURF as a keypoint detector and RootSIFT as a features descriptor. Thus, fusing the FAST and SURF keypoints is suggested by selecting the unique keypoints. Then, the RootSIFT was used to describe those keypoints.

The FAST and SURF keypoints were detected separately from each image in the dataset. Then, they were concatenated uniquely to prevent any point from duplication. Then the RootSIFT describes the detected keypoints. If an inquiry example image is given, its quantized features histogram will be predicted by the trained SVMs model. Moreover, it will be compared with the quantized features histograms of the images in the target dataset based on the Cosine distance similarity metrics. The result of the Top-20 retrieved images in response to the queries belonging to the Elephants class is shown in Figure 10 The query image appears in the first row. In contrast, the other two rows are the retrieved images in response to the query image. The score of each extracted image is displayed above it, and it represents the position of the query image to the retrieved image.

In Figure 11 and Figure 12, the various vocabulary sizes are shown along with the X-axis and the MAP value on the Y-axis. The Blue and Red lines show the CBIR performance of the standalone FAST and SURF keypoint based on the BoVW technique. The third line shows the proposed KFFVW framework’s performance, giving the max MAP value of 92.49% at vocabulary size 8192. Furthermore, the MAP value of 91.19% on a vocabulary size of 2048 using 50% of the features, as shown in Figure 11 at using 75% features with 8192 vocabulary sizes, the MAP value is 91.59%, as shown in Figure 12.

Table 4 shows the performance and statistical analysis of the suggested HKFV+BFS approach based on the fusion FAST and SURF keypoint detectors on various sizes of the vocabulary and BFS percentages using the Corel-1000 image dataset. The bold value indicates the best MAP performance. As shown in Table 1, the overall calculation time required for the whole retrieval process for a CBIR in the proposed work is divided into feature extraction time, prediction time, and similarity measurement and sorting time. The comparison between the proposed framework and the recent CBIR approaches (GMM-EM [45], Hybrid features [46], CH-LDP [47], CH-LDP-SIFT-DPBoF [48], and GA-SVM [49]) in Table 5 proves the adequate performance of the suggested method. Furthermore, according to the same table, the proposed framework outperforms all other methods in the mean average precision. The suggested framework surpasses the other retrieval results in most of the classes.

#### 4.3.2. Caltech-10 Dataset Results

The Caltech-10 dataset consists of 1000 images of 10 semantic classes, each containing 100 images. It is a subgroup of the Caltech-101 dataset, which includes 101 classes. In our work, the classes Faces, Leopards, Motorbikes, Airplanes, Bonsai, Brain, Car side, Grand piano, Ketch, and Watch were selected from the Caltech-101 dataset to construct the Caltech-10 dataset. To investigate the effect of the combination of BFS with the vocabulary size on the performance of the proposed framework, we have designed a combination between BFS and vocabulary size as mentioned in Table 6. The best precision value (i.e., 98.40%) was at the combination 75% for BFS and 2048 vocabularies. Figure 13 shows the combination of the BFS and the vocabulary size on the Caltech-10 dataset. The results in Table 7 show that the proposed KFFVW framework based on HKFV features using BFS gives better results than using a standalone keypoint feature detector such as SURF and FAST.

#### 4.3.3. Flower-17 Dataset Results

We computed the performance via various experiments of the proposed framework on the Oxford Flower-17 dataset on different combinations of BFS and vocabulary sizes. The results in Table 8 show the effectiveness of the HKFV+BFS techniques on various vocabulary sizes and BFS. The best MAP value is 84.94% at the combination of 25% BFS and 8192 vocabularies. Figure 14 shows the relation between the percentages of BFS and the vocabulary size. It also shows that the low and high percentages of BFS may lead to data overfitting or low results, while the appropriate values for BFS and vocabulary size (i.e., 25% and 8192, respectively) give the best results. Furthermore, the traditional BoVW technique experimented on the Oxford Flower-17 dataset using standalone SURF and standalone FAST keypoint features detector with the RootSIFT feature descriptor. Table 9 shows that the proposed KFFVW framework based on HKFV features and BFS visual words gives better results than a standalone keypoint feature detector.

In this experiment, we evaluated how well our framework performed using various numbers of images it returned. The classes of Sunflower, Pansy, Coltsfoot, Windflower, Tulip, Lilyvalley, Iris, Fritillary, and Buttercup yielded the highest results. In contrast, the classes Crocus, Bluebell, Cowslip, Daffodil, Dandelion, Snowdrop, Tigerlily, and Daisy yielded the lowest results because they have similar features and less dissimilarity over other semantic classes. The proposed framework attained a maximum mean average precision of 84.94% on the Oxford Flower-17 dataset. The comparison between the proposed framework and the recent CBIR approaches [50,51,52] on the Oxford Flower-17 dataset in Table 10 proves the adequate performance of the suggested method.

It was interesting to find the best combination between SIFT, FAST, SURF, and ORB to produce the HKFV. The performance and execution time comparison results between HKFV(SIFT+ ORB)+BFS, HKFV(SIFT+ORB+ FAST+SURF)+BFS, and HKFV(FAST+ SURF)+BFS were included as Table 2. The comparison in Table 3 provides execution time and performance between (SURF+BFS), (FAST+BFS), (HKFV), and (HKFV+BFS) validates the effectiveness of each component in the framework and indicates whether each part is necessary for the framework.

The tests and computations were performed on an AMD Ryzen 5 3400G with Radeon Vega 11 graphics (4 CPUs) 3.7 GHz, RAM 8 GB PC running Ubuntu 18.04 LTS 64bit.

## 5. Conclusions and Future Works

This paper has presented a novel CBIR framework based on visual word fusion of multiple feature descriptors to improve retrieval performance. The interest points have been separately extracted from an image using FAST and SURF. The retrieved keypoints have been combined into one keypoint feature vector, and the enhanced RootSIFT technique has been used to describe the area around each keypoint. The FeatureWiz selection technique has been applied to reduce and select the robust features for the BoVW learning model. K-means clustering has been used to quantify visual features into a more manageable set of visual words to produce the codebook. Finally, The SVM classifier for image retrieval has been fed by the feature vectors generated from the BoVW model. The obtained images have been sorted according to how closely they resemble the query image using an inverted index technique based on the cosine distance metric. By achieving average accuracies of 92.94%, 98.40%, and 84.94% on these benchmark datasets (Corel-1000, Caltech-10, and Oxford Flower-17), experiments on the three benchmark datasets have demonstrated that the presented CBIR technique could deliver results that were comparable to other state-of-the-art methods. Nevertheless, a possible limitation of this study is that the validation process needs wider datasets. The future work will be in two main directions. The first one is applying advanced deep learning techniques for better retrieval similarity and accuracy as an adaptive version of the framework. The second direction will focus on performing more experiments for testing and evaluating our method for addressing content-based image retrieval systems and presenting it as an online API.

## Figures and Tables

**Figure 1 sensors-23-01653-f001:**
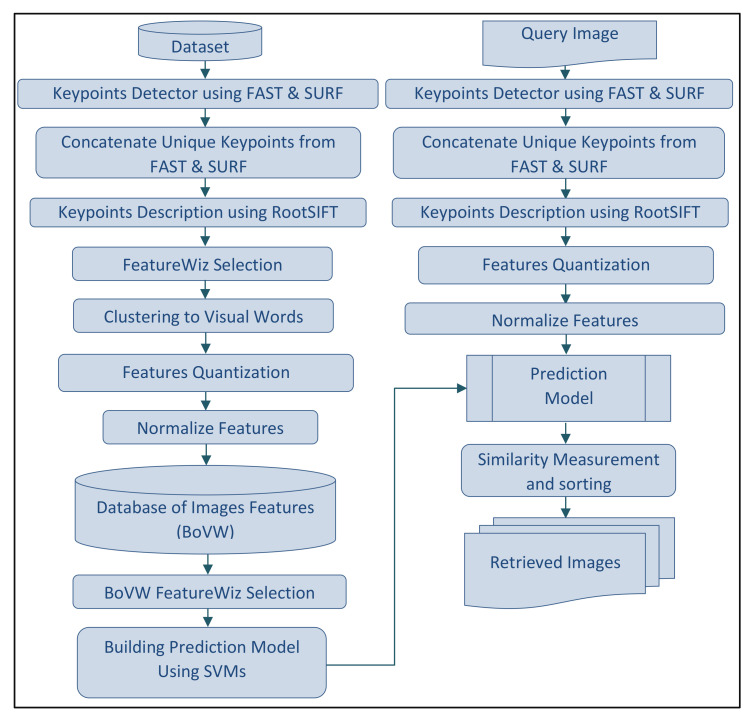
Flowchart of the proposed methodology.

**Figure 2 sensors-23-01653-f002:**
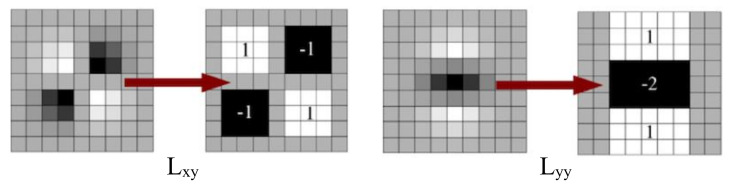
Box filter approximation of second−order Gaussian partial derivatives.

**Figure 3 sensors-23-01653-f003:**
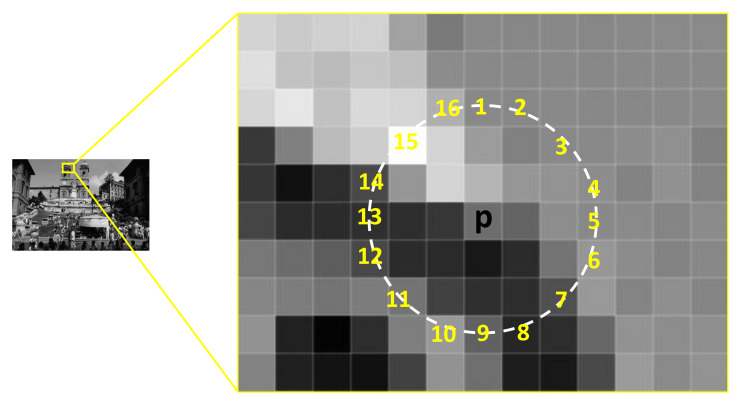
An interest point and 16 pixels surrounding on it.

**Figure 4 sensors-23-01653-f004:**
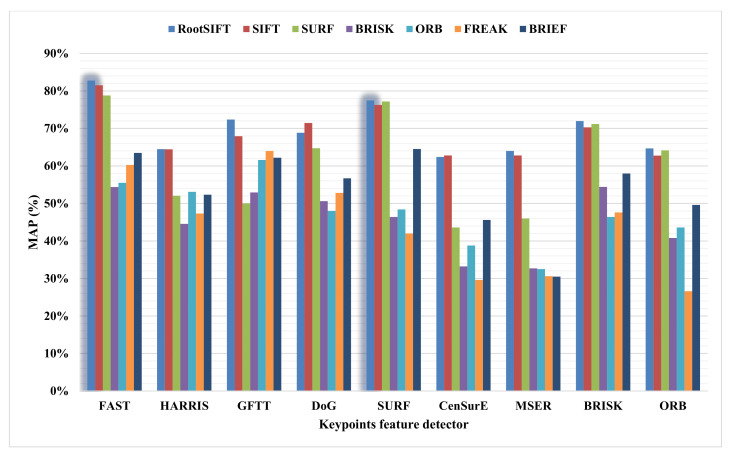
Performance of each keypoint detector with each descriptor.

**Figure 5 sensors-23-01653-f005:**
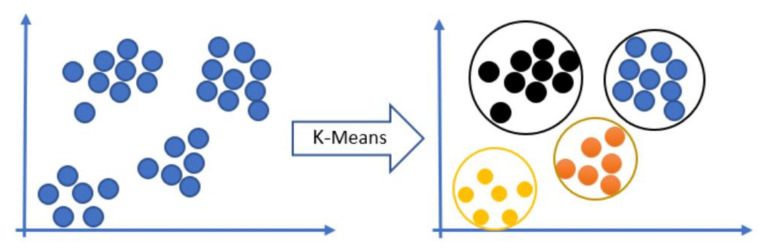
K-Means clustering model sample.

**Figure 6 sensors-23-01653-f006:**
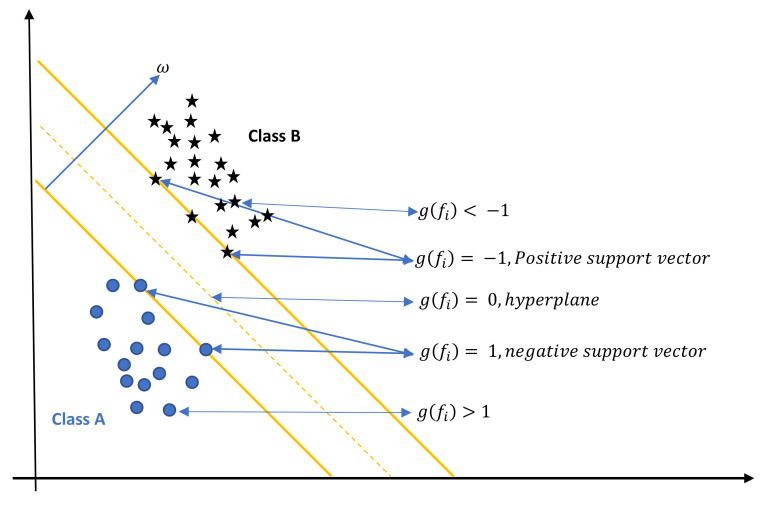
SVMs maximize the hyperplane between two different classes, A and B.

**Figure 7 sensors-23-01653-f007:**
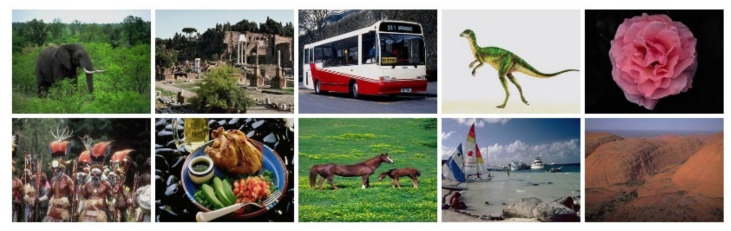
Sample image of each semantic category of the Corel-1000 dataset.

**Figure 8 sensors-23-01653-f008:**
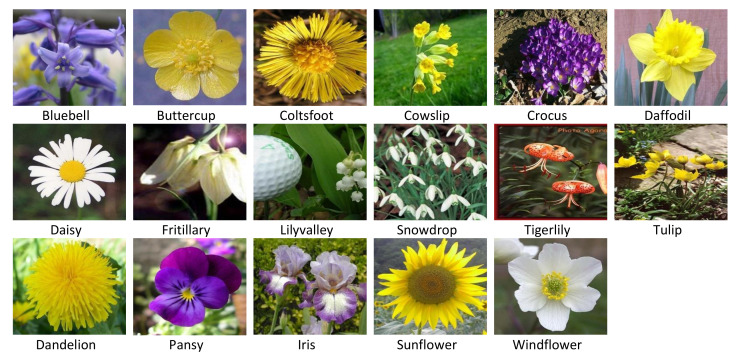
Sample image of each semantic category of the Caltech-10 dataset.

**Figure 9 sensors-23-01653-f009:**
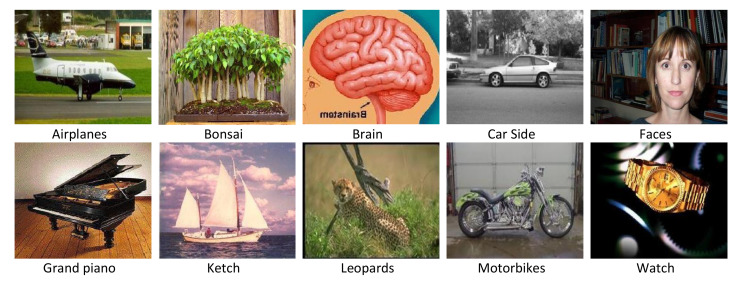
Sample image of each semantic category of Oxford Flower-17 dataset.

**Figure 10 sensors-23-01653-f010:**
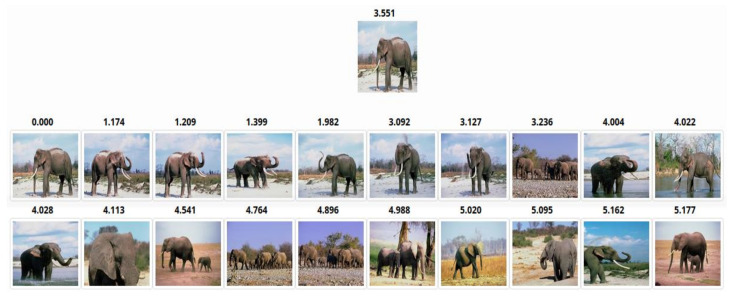
The retrieved images for an Elephant image from the Corel-1000 dataset.

**Figure 11 sensors-23-01653-f011:**
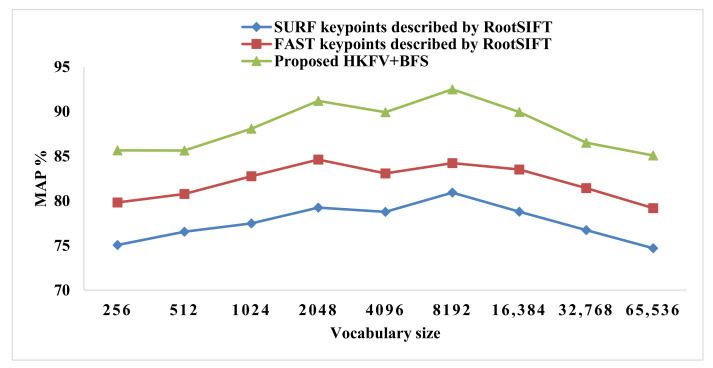
HKFV using 50% BFS vs. standalone FAST and SURF on the Corel-1000 dataset.

**Figure 12 sensors-23-01653-f012:**
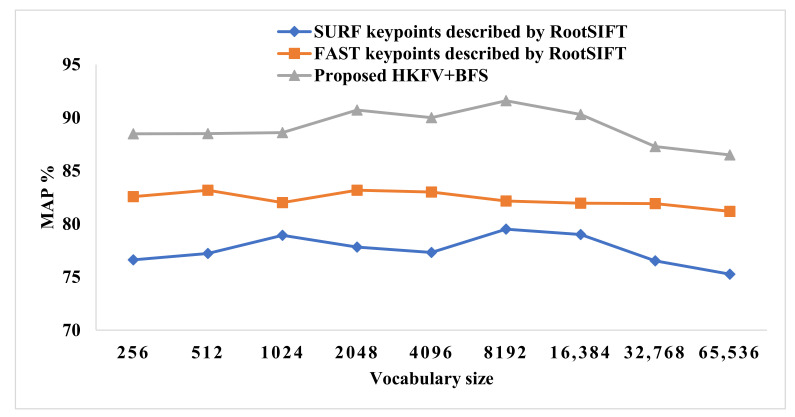
HKFV using 75% BFS vs. FAST and SURF keypoints detector on the Corel-1000 dataset.

**Figure 13 sensors-23-01653-f013:**
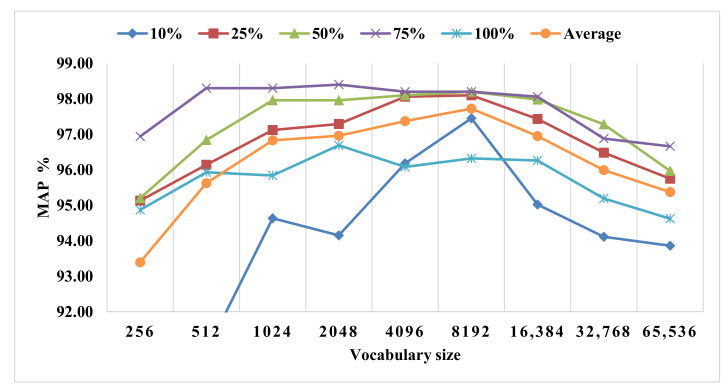
KFFVW on various vocabularies sizes and BFS on the Caltech -10 dataset.

**Figure 14 sensors-23-01653-f014:**
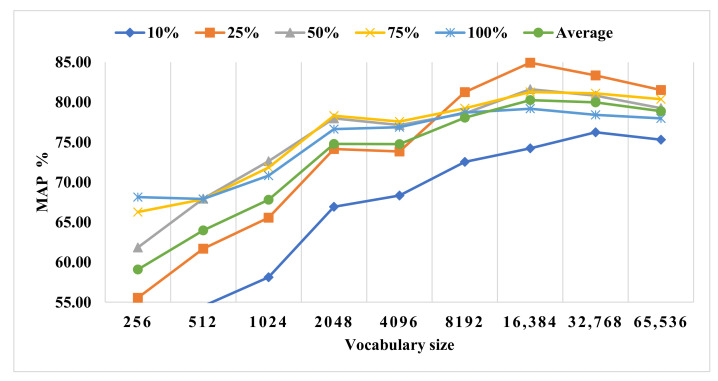
KFFVW on various vocabularies sizes and BFS on Oxford Flower-17 dataset.

**Table 1 sensors-23-01653-t001:** Time spent for each phase of the retrieval process in seconds.

Phase	Elapsed Time (s)
Query Feature Extraction (QFE)	0.140
Query Prediction (QP)	0.001
Similarity and Sorting of top 20 Images (SaS20)	0.007
Total	0.148

**Table 2 sensors-23-01653-t002:** Various features combination performance for the proposed method on Oxford Flower-17 dataset.

	HKFV (SIFT+ORB) +BFS	HKFV (SIFT+ORB+FAST+SURF) +BFS	HKFV (FAST+SURF) +BFS
MAP	81.72	82.86	84.94
QFE Time	0.1281	0.2618	0.1401
QP Time	0.0010	0.0014	0.0011
SaS20 Time	0.0070	0.0078	0.0072
Total Time	0.1361	0.271	0.1484

**Table 3 sensors-23-01653-t003:** Comparison validate the effectiveness of each component in the framework.

	SURF	SURF+BFS	FAST	FAST+BFS	HKFV	HKFV+BFS
MAP	74.88	78.13	79.06	81.23	82.95	84.94
QFE Time	0.0652	0.0706	0.0531	0.0586	0.1294	0.1401
QP Time	0.0009	0.0010	0.0007	0.0008	0.0010	0.0011
SaS20 Time	0.0053	0.0058	0.0062	0.0065	0.0070	0.0072
Total Time	0.0714	0.0774	0.0600	0.0659	0.1374	0.1484

**Table 4 sensors-23-01653-t004:** The KFFVW Performance with various vocabulary sizes and BFS on the Corel-1000 dataset.

F%	MAP Performance (in %) on Different Vocabulary Size								
	256	512	1024	2048	4096	8192	16,384	32,768	65,536
10%	60.36	59.74	75.88	82.41	88.55	87.35	87.18	83.80	77.57
25%	82.24	81.63	85.88	87.50	88.55	91.60	90.34	89.54	88.34
50%	85.65	85.63	88.08	91.19	89.92	**92.49**	89.95	86.50	85.08
75%	88.47	88.49	88.60	90.72	90.00	91.59	90.29	87.28	86.49
100%	86.68	87.06	87.73	89.74	89.40	90.26	89.01	84.77	83.59
Average	80.88	80.71	85.43	88.51	89.48	90.85	89.55	86.58	84.21
SE	5.18	5.32	2.38	1.61	0.32	0.90	0.59	1.00	1.84
SD	11.58	11.89	5.33	3.59	0.71	2.01	1.33	2.24	4.11

**Table 5 sensors-23-01653-t005:** Comparison between metrics the recent proposed CBIR approaches on the Corel-1000 and the proposed method.

Classes	Metrics	GMM-EM [45]	Hybrid Features [46]	CH-LDP [47]	CH-LDP-SIFT-DPBoF [48]	GA-SVM [49]	Proposed HKFV+BFS
	P	72.50	79.90	77.9	82.60	85.00	83.57
Africa	R	14.50	15.98	15.80	16.52	17.00	18.64
	F	24.17	26.63	26.27	27.53	28.33	30.48
	P	65.20	47.25	60.10	56.80	85.00	88.22
Beach	R	13.40	9.45	12.20	11.36	17.00	15.64
	F	22.23	15.75	20.28	18.93	28.33	26.57
	P	70.60	65.70	69.10	77.10	85.00	94.63
Building	R	14.12	13.14	13.82	15.42	17.00	19.78
	F	23.53	21.90	23.03	25.70	28.33	32.72
	P	89.20	92.95	87.60	98.90	95.00	100.0
Buses	R	17.84	18.59	17.52	19.78	19.00	20.00
	F	29.73	30.98	29.20	32.97	31.67	33.33
	P	100.0	99.85	99.40	100.0	100.0	100.0
Dinosaurs	R	20.00	19.97	19.88	20.00	20.00	20.00
	F	33.33	33.28	33.13	33.33	33.33	33.33
	P	70.50	63.60	59.25	75.50	100.0	96.31
Elephants	R	14.10	12.72	11.85	15.10	20.00	19.54
	F	23.50	21.20	19.75	25.17	33.33	32.49
	P	94.80	93.35	95.80	97.30	95.00	93.08
Flowers	R	18.96	18.67	19.16	19.46	19.00	18.86
	F	31.60	31.12	31.93	32.43	31.67	31.36
	P	91.80	95.55	91.85	95.90	95.00	91.85
Horses	R	18.36	19.11	18.37	19.18	19.00	18.20
	F	30.60	31.85	30.62	31.97	31.67	30.38
	P	72.25	41.60	64.00	77.80	85.00	93.12
Mountains	R	14.45	8.32	12.80	15.56	17.00	19.62
	F	24.08	13.87	21.33	25.93	28.33	32.41
	P	78.80	76.15	78.10	89.50	85.00	84.15
Foods	R	15.76	15.23	15.62	17.90	17.00	18.25
	F	26.27	25.38	26.03	29.83	28.33	29.99
MAP		80.57	75.59	78.31	85.14	91.00	**92.49**
Avg-Recall		16.15	15.12	15.70	17.03	18.20	18.85
Avg-F-score		26.85	25.19	26.16	28.37	30.33	31.31

**Table 6 sensors-23-01653-t006:** The KFFVW Performance with various vocabulary sizes and BFS on the Caltech-10 dataset.

BFS%	MAP Caltech-10 Dataset Performance (in %) on Different Vocabulary Size								
	256	512	1024	2048	4096	8192	16384	32,768	65,536
10%	84.82	90.87	94.63	94.15	96.18	97.45	95.02	94.11	93.86
25%	95.13	96.14	97.12	97.29	98.06	98.10	97.43	96.48	95.75
50%	95.20	96.84	97.96	97.96	98.10	98.20	97.98	97.28	95.97
75%	96.94	98.30	98.30	**98.40**	98.20	98.20	98.06	96.88	96.66
100%	94.87	95.93	95.84	96.69	96.08	96.32	96.26	95.19	94.62
Average	93.39	95.62	96.83	96.96	97.37	97.72	96.95	95.99	95.37
SE	2.174	1.257	0.683	0.746	0.488	0.362	0.580	0.586	0.501
SD	4.862	2.811	1.527	1.668	1.092	0.809	1.297	1.311	1.119

**Table 7 sensors-23-01653-t007:** HKFV with BFS vs. standalone SURF and FAST on Caltech-10 dataset classes.

	SURF			FAST			Proposed HKFV+BFS		
**Classes**	**P**	**R**	**F**	**P**	**R**	**F**	**P**	**R**	**F**
Faces	98	20	33	100	20	33	100	20	33
Leopards	96	19	32	98	20	33	100	20	33
Motorbikes	92	18	31	93	19	31	97	19	32
Airplanes	92	18	31	93	19	31	97	19	32
Bonsai	88	18	29	88	18	29	100	20	33
Brain	88	18	29	90	18	30	97	19	32
Car side	88	18	29	90	18	30	96	19	32
Grand piano	91	18	30	93	19	31	97	19	32
Ketch	91	18	30	92	18	31	100	20	33
Watch	91	18	30	92	18	31	100	20	33
Average	91.50	18.30	30.50	92.90	18.58	30.97	**98.40**	**19.68**	**32.80**

**Table 8 sensors-23-01653-t008:** The KFFVW Performance with various vocabulary sizes and BFS on Oxford Flower-17.

BFS%	MAP Flower-17 Dataset Performance (in %) on Different Vocabulary Size								
	256	512	1024	2048	4096	8192	16384	32,768	65,536
10%	43.63	54.50	58.12	66.92	68.34	72.54	74.23	76.24	75.32
25%	55.55	61.68	65.56	74.14	73.83	81.26	**84.94**	83.34	81.51
50%	61.84	67.94	72.63	77.98	77.14	78.59	81.62	80.82	79.25
75%	66.27	67.86	71.84	78.30	77.57	79.23	81.27	81.11	80.37
100%	68.13	67.90	70.83	76.62	76.88	78.69	79.18	78.42	77.98
Average	59.08	63.98	67.80	74.79	74.75	78.06	80.26	79.99	78.89
SE	4.43	2.66	2.72	2.10	1.73	1.46	1.76	1.22	1.07
SD	9.90	5.94	6.07	4.70	3.88	3.27	3.95	2.72	2.39

**Table 9 sensors-23-01653-t009:** HKFV with BFS vs. standalone SURF and FAST on Oxford Flower-17 dataset.

	SURF			FAST			Proposed HKFV+BFS		
**Classes**	**P**	**R**	**F**	**P**	**R**	**F**	**P**	**R**	**F**
Bluebell	76	15.2	25	74	15	25	80	16	27
Buttercup	70	14	23	82	16	27	85	17	28
Coltsfoot	77	15.4	26	85	17	28	93	19	31
Cowslip	64	12.8	21	72	14	24	80	16	27
Crocus	74	14.8	25	73	15	24	78	16	26
Daffodil	64	12.8	21	76	15	25	80	16	27
Daisy	74	14.8	25	75	15	25	84	17	28
Dandelion	70	14	23	71	14	24	80	16	27
Fritillary	69	13.8	23	84	17	28	85	17	28
Iris	82	16.4	27	84	17	28	85	17	28
Lilyvalley	70	14	23	79	16	26	85	17	28
Pansy	87	17.4	29	86	17	29	93	19	31
Snowdrop	67	13.4	22	72	14	24	80	16	27
Sunflower	94	18.8	31	94	19	31	100	20	33
Tigerlily	72	14.4	24	75	15	25	80	16	27
Tulip	78	15.6	26	83	17	28	88	18	29
Windflower	85	17	28	79	16	26	88	18	29
Average	74.88	14.98	24.96	79.06	15.81	26.35	**84.94**	16.99	28.31

**Table 10 sensors-23-01653-t010:** Comparison between recent CBIR approaches and the HKFV+BFS on Oxford Flower-17 dataset.

Method/Metrics	P	R	F
Color and texture [50]	76.4	15.3	25.49
Elalami [51]	79.3	16.1	26.77
Elalami [52]	82.1	17.3	28.58
Proposed Framwork	**84.94**	16.99	28.31

## Data Availability

The data presented in this study are available on request from the corresponding authors.

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
