# Peer review of "Hybrid Bag-of-Visual-Words and FeatureWiz Selection for Content-Based Visual Information Retrieval"

_sensors, 2023, doi:10.3390/s23031653_

Round 1
Reviewer 1 Report
In this paper, we propose a new strategy for combining FAST and SURF features, which is called the Hybrid Keypoint Feature Vector. In the experiment, the proposed method resulted in better accuracy than state of the art. Even though the idea is quite interesting but there is no scientific explanation why what is combined is FAST and SURF? Actually besides FAST and SURF there is a keypoint extractor that has good accuracy, namely ORB. Why not use ORB? It will be very interesting if the author finds the best combination between SIFT, FAST, SURF and ORB to produce a Hybrid Keypoint Feature Vector.
Several suggestion for the authors
- The author need to elaborate the major contribution of their work in the introduction/related works part.
- Mathematically, how do you performing to select unique keypoints concatenating from FAST and SURF?
- Why are the keypoints represented by RootSIFT?
- Is there any comparison results between FAST+SURF and other keypoint combination in the experiment?
- What about the processing time required? It would be very interesting if there was a discussion regarding this.
Reviewer 2 Report
The results obtained are very good for the test set used. I believe that images in which the contrast between the subject and the background is not to high should be introduced in the tests. In the test images presented in the paper, the objects, that constitute the subject, are too "obvious".
Author Response
We thank the reviewer for accepting our manuscript. Also we thank him for his time in reading and for providing valuable suggestions that can help advance the level of our manuscript.
Reviewer 3 Report
The authors in this paper proposed a framework based on hybrid features and a feature selection for content-based visual information retrieval (CBIR) tasks. Intensive studies have been conducted to validate the effectiveness of the proposed framework. The paper writing is very clear. However, there is a lack of ablation studies to validate the effectiveness of each component in the framework (hybrid features and feature selection). Thus it is not clear whether each part is necessary for the framework. Here are the comments about the paper.
- Can the authors compare the results related to HKFV + BFS with those related to SURF + BFS and those related to FAST + BFS?
- Can the authors also validate the effectiveness of BFS by comparing the results related to HKFV (without BFS) and those related to HKFV + BFS? Also, can other feature selection methods replace the BFS here?
- Can the framework be generalized to a different prediction model?
- The performance metrics ‘MAP’ was not defined.
